# HIV Infection Indicator Disease-Based Active Case Finding in a University Hospital: Results from the SHOT Project

Andrea De Vito [1],*, Agnese Colpani [1], Maria Sabrina Mameli [1], Paola Bagella [1], Vito Fiore [1], Claudio Fozza [2], Maria Antonia Montesu [3], Alessandro Giuseppe Fois [4], Fabiana Filigheddu [5], Noemi Manzoni [6], Carlo Putzu [7], Sergio Babudieri [1] and Giordano Madeddu [1]

1  Unit of Infectious Diseases, Department of Medicine, Surgery and Pharmacy, University of Sassari, 07100 Sassari, Italy
2  Unit of Haematology, Department of Medicine, Surgery and Pharmacy, University of Sassari, 07100 Sassari, Italy
3  Unit of Dermatology, Department of Medicine, Surgery and Pharmacy, University of Sassari, 07100 Sassari, Italy
4  Unit of Respiratory Diseases, Department of Medicine, Surgery and Pharmacy, University of Sassari, 07100 Sassari, Italy
5  Unit of Internal Medicine, Department of Medicine, Surgery and Pharmacy, University of Sassari, 07100 Sassari, Italy
6  Unit of Internal Medicine, University Hospital of Sassari, 07100 Sassari, Italy
7  Unit of Oncology, University Hospital of Sassari, 07100 Sassari, Italy
*  Correspondence: andreadevitoaho@gmail.com; Tel.: +39-340-470-4834

**Abstract:** In 2014, UNAIDS launched renewed global targets for HIV control to achieve by 2025, known as "the three 95": 95% of people living with HIV (PWH) diagnosed, of which 95% are receiving treatment, of which 95% are on sustained virological suppression. In Italy, new HIV diagnoses have been steadily decreasing since 2012. However, in 2020, 41% of new diagnoses presented with less than 200 CD4+ cells/μL and 60% with less than 350 CD4+ cells/μL. Implementing testing and early treatment is a key strategy to prevent AIDS, late presentation, and HIV transmission. We selected non-Infectious Diseases Units based on the European project HIDES and engaged colleagues in a condition-guided HIV screening strategy. We enrolled 300 patients, of which 202 were males (67.3%) and 98 were females (32.7%). Most of the screening was performed in Infectious Diseases (ID) and Hematologic wards. In total, we diagnosed eleven new HIV infections with a hospital prevalence in the study population of 3.7%. Five (45.4%) had a CD4 count $<100/mm^3$, one (9.1%) $<200/mm^3$, and one (9.1%) $<300/mm^3$. Regarding risk factors, 81.8% declared having had unprotected sexual intercourse and 54.5% were heterosexual. All patients promptly started a combination antiretroviral regimen and 10 (90.9%) obtained an undetectable HIV-RNA status. Eight of the eleven (72.7%) patients are currently on follow-up in our outpatient clinic. A proactive indicator disease-guided screening can help avoid missed opportunities to diagnose HIV infection in a hospital setting. Implementing this kind of intervention could favor early diagnosis and access to treatment.

**Keywords:** HIV; AIDS; testing strategies; active case finding; indicator disease

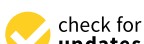



## 1. Introduction

Human immunodeficiency virus infection and acquired immunodeficiency syndrome (HIV/AIDS) were first described in 1981. Since then, the pandemic has shifted from a rising trend in the 1980s and 1990s to a declining number of new diagnoses from the 2000s [1]. However, HIV remains a major public health issue worldwide, with around 37.7 million people living with HIV in 2020 [2].

Common symptoms of acute HIV infection overlap with a mononucleosis-like syndrome. Afterwards, patients can remain asymptomatic for years or present generic symptoms such as lymphadenopathy, diarrhea, fatigue, fever, sweating, oral candidiasis,

and herpes zoster. Then, the disease can evolve into an AIDS-defining condition (opportunistic infections, certain cancer variants, neurological disorders) [3].

Even though HIV is not curable, early diagnosis and treatment have shown to be effective measures to improve immunological recovery and quality of life and reduce disease progression and HIV transmission [4]. Through PARTNER and PARTNER2 studies, Rodger et al. demonstrated that zero new transmission occurs when people living with HIV (PWH) on antiretroviral treatment (ART) and are virologically suppressed have condomless sex with their partners [5–9]. Furthermore, thanks to the new regimens, most PWH achieve an undetectable HIV-RNA status in the first year of treatment, with a low incidence of virological failure and adverse events [4,10,11]. This reinforces the importance of early and rapid ART initiation. Nonetheless, UNAIDS advocates 95% of PWH knowing their status, 95% being on diagnosis ART, and 95% being virologically suppressed by 2025 [12]. Yet, in 2020, only 84% of PWH knew their HIV status, with around 6 million people not knowing they were living with HIV. In addition, only 73% of PWH were accessing ART in 2020, and 66% were virologically suppressed [13]. Hence, one of the key issues to address is how to diagnose more PWH at an early stage of the infection.

In Italy, new HIV diagnoses have been steadily decreasing since 2012. However, in 2020, 41% of new diagnoses presented with less than 200 CD4+ cells/μL and 60% with less than 350 CD4+ cells/μL [14]. Implementing testing and early treatment is a key strategy to prevent AIDS and late presentation. HIV testing is the first crucial step in the HIV cascade of care. To improve that, stigmatization must be addressed to encourage people to undergo routine testing for HIV, as cultural norms and stigma often represent barriers to testing access [15]. In addition, community-based interventions must be promoted, although some progress has already been made in this area with the creation of fast-track cities [16].

Moreover, health workers must be sensitized and trained to inform patients properly and offer the test when appropriate in the hospital setting. Unfortunately, few data regarding non-Infectious Diseases health workers' attitudes towards HIV testing are available. A recently published study from Australia highlighted perceived stigma, testing and results responsibility, and limited resources as the main reasons for testing hesitancy [17]. The European guidelines for HIV testing recommend the normalization of HIV testing in the hospital setting, and one of the strategies proposed is the indicators of condition-guided testing [18]. In this regard, we promoted a facility-based intervention to encourage hospital-based screening when an HIV indicator condition is present. This project was inspired by the HIDES (HIV-Indicator Diseases across Europe Study), whose conclusions encourage reinforcing facility-based indicator condition-guided HIV screening. Our objectives were to give patients the opportunity to know their serological status, sensitize colleagues about suspecting HIV, be proactive in offering the test, and normalize the diagnosis.

## 2. Materials and Methods

We conducted a prospective observational study called the SHOT Project (Screening HIV Ospedale Territorio). The study was conducted from April 2017 to October 2018 at the University Hospital of Sassari, Italy. The units of Infectious Diseases (ID), Hematology, Oncology (ON), Respiratory Diseases (RD), Dermatology, and Internal Medicine (IM) were engaged in the study. The patients eligible for screening were defined as patients presenting with at least one of the following conditions: (i) AIDS-defining conditions; (ii) conditions associated with an undiagnosed HIV prevalence of at least 0.1% (individuals presenting with these conditions when tested for HIV have a positive testing rate of at least 1/1000); (iii) not identifying the presence of an HIV infection may have significant adverse implications for the individual's clinical management (e.g., for conditions requiring chemotherapy or biologics) [18].

Each ward was provided with a dedicated form to collect demographical and epidemiological data from the patients included in the screening. In addition, each patient signed a written informed consent to participate in the study and undergo HIV screening.

The screening was performed through enzyme-linked immunosorbent assay (ELISA) on serum samples. If positive, a confirmation test through immunoblotting and HIV-RNA real-time polymerase chain reaction (RT-PCR) was performed. At this point, patients were taken into charge by the Unit of ID.

*Ethical Issues*

The study was conducted in accordance with the Helsinki Declaration approved in 2003, and the protocol was approved by the local ethic committee (Protocol 248I/CE, 11 April 2017).

## 3. Results

We enrolled 300 patients, of which 202 were males (67.3%) and 98 were females (32.7%). The median age was 41 (26–58) years old. Most of the screening was performed in the Infectious Diseases and Hematologic wards (Figure 1).

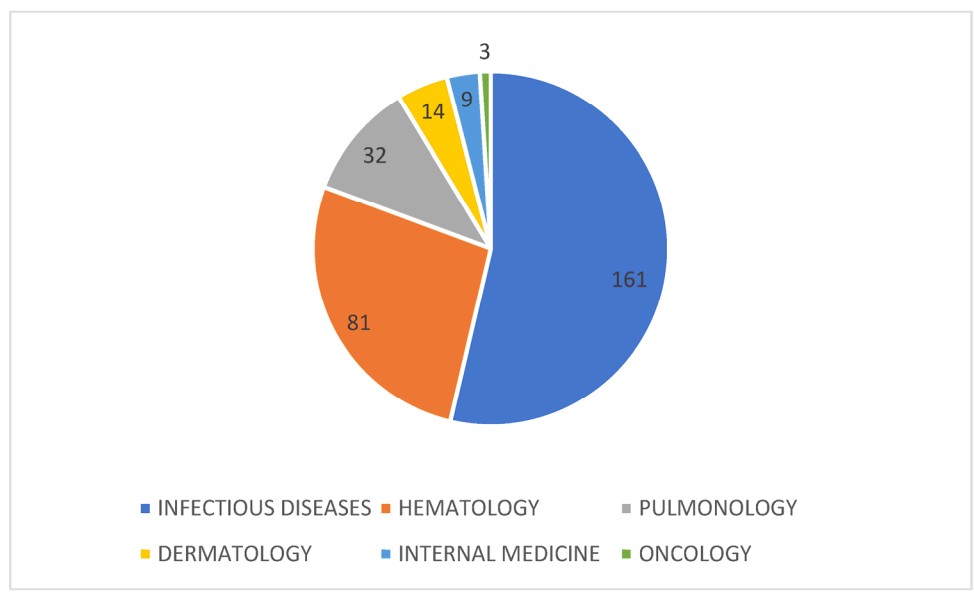

**Figure 1.** Distribution of the 300 HIV screenings among the different wards.

The main reasons for screening were fever, lymphadenopathy, non-Hodgkin lymphoma, and sexually transmitted diseases (Figure 2).

In total, we diagnosed eleven new HIV infections with a hospital prevalence in the study population of 3.7%. Five patients (45%) were diagnosed in the RD Unit, with a positivity rate of 5/32 (15.6%). Four patients (36%) were tested and analyzed in ID, with a positivity rate of 2.5% in our ward, and two (19%) in IM, with a positivity rate of 22.2%. Of these, five (45.4%) had a CD4 count $<100/\text{mm}^3$, one (9.1%) $<200/\text{mm}^3$, one (9.1%) $<300/\text{mm}^3$, two (18.2%) $>300/\text{mm}^3$, and two (18.2%) $>900 \text{ mm}^3$. Eight (72.7%) people had a viral load >100.000 cp/mL. Among AIDS presenters, the CDC stage was C3 in five (45.4%). Regarding risk factors, 81.8% declared having had unprotected sexual intercourse and 54.5% were heterosexual. The characteristics of the patients are reported in Table 1.

All patients promptly started a combination antiretroviral regimen and obtained an undetectable HIV RNA status. Eight out of the eleven patients are currently on follow-up in our outpatient clinic. Unfortunately, one patient was lost to follow-up, and one patient died. The remaining patient moved to another country.

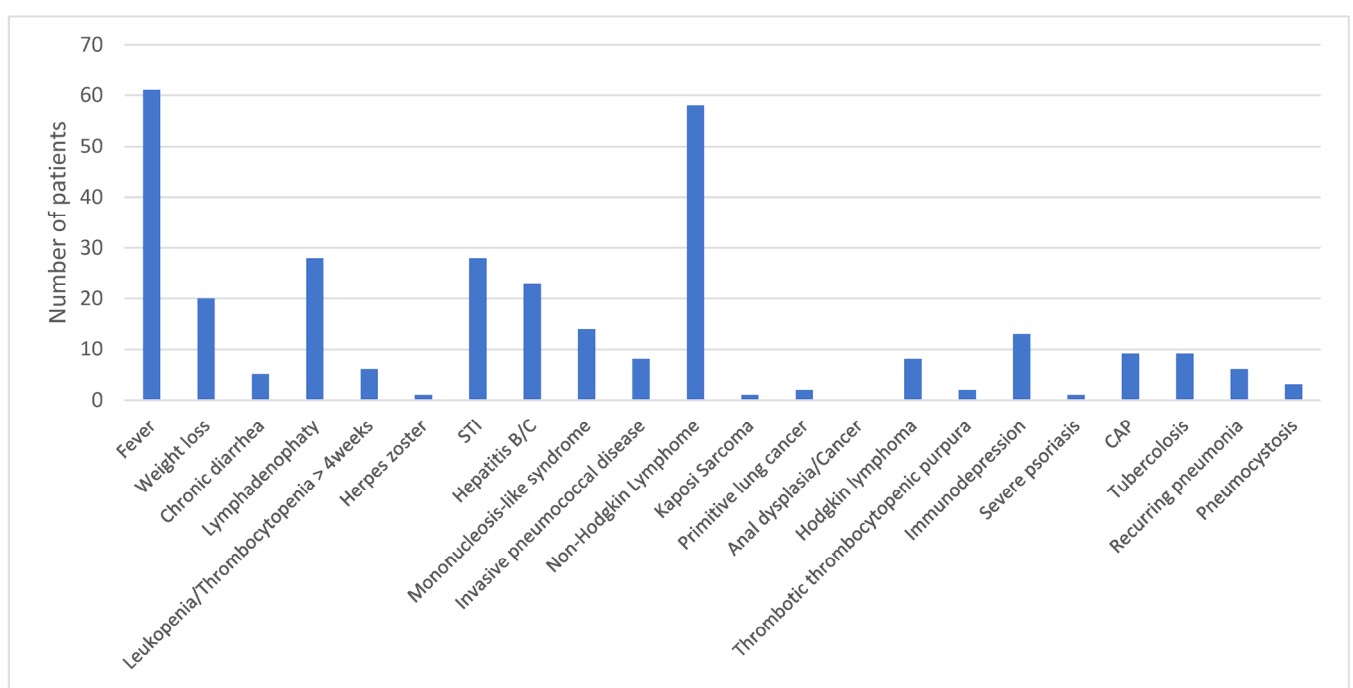

**Figure 2.** Reasons why HIV screening was proposed to the 300 patients included in the study. STI: sexually transmitted infection; CAP: community-acquired pneumonia.

**Table 1.** Characteristics of 11 patients diagnosed with HIV in the framework of the SHOT Project in Sassari, Italy, from April 2017 until October 2018.

| Patient | Sex | Age | Risk Factor | Reason for Testing | Opportunistic Infections | Stage | Basal HIV-RNA (Copies/mL) | CD4 Cells/mm³ | CD4 % | CD4/CD8 | First-Line Therapy | Date of Negativization | LTFU |
|---|---|---|---|---|---|---|---|---|---|---|---|---|---|
| 1 | Female | 29 | HE | Fever | None | B3 | 2,000,000 | 54.00 | 5.62 | 0.12 | TDF/FTC + ATV/r | 10 February 2018 | Yes |
| 2 | Female | 41 | HE | Pharyngotonsillitis; Rash | None | A1 | 9420 | 1357.00 | 52.15 | 1.63 | TDF/TAF + DTG | 27 September 2019 | No |
| 3 | Male | 43 | MSM | Bilateral pneumonia | PjP | C3 | 6480 | 56.00 | 3.14 | 0.04 | TDF/FTC + DTG | 8 November 2017 | No |
| 4 | Female | 42 | HE | Bilateral pneumonia | PjP | C3 | 1,940,000 | 80.00 | 5.2 | 0.19 | TDF/FTC + DTG | Died | No |
| 5 | Male | 40 | MSM | Bilateral pneumonia | PjP | A2 | 150,000 | 308.00 | 23.00 | 0.42 | TDF/FTC + DTG | 25 July 2017 | No |
| 6 | Male | 40 | IDU | Bilateral pneumonia | PjP | C3 | 739,000 | 14.00 | 0.48 | 0.01 | TDF/FTC + DTG | 17 July 2017 | No |
| 7 | Female | 57 | HE | Pleural Empyema | None | B2 | 139,000 | 317.00 | 10.72 | 0.14 | TAF/FTC + RAL | 21 March 2019 | No |
| 8 | Male | 72 | HE | Fever; Asthenia | HZV | C3 | 462,000 | 162.00 | 17.97 | 0.62 | TDF/FTC + RAL | 16 March 2018 | No |
| 9 | Male | 66 | HE | Fever; Asthenia | None | B1 | 765,000 | 924.00 | 9.77 | 1.39 | TDF/FTC + DRV/r | 12 September 2017 | Yes |
| 10 | Male | 28 | MSM | Bilateral pneumonia | PCP; Invasive Pneumococcal Disease; CMV; Esophageal Candidiasis; Perianal HSV | C3 | 150,000 | 29.00 | 3.89 | 0.07 | TDF/FTC + DTG | 1 June 2017 | No |
| 11 | Male | 73 | MSM | Stomatitis | None | B2 | 73,100 | 246.00 | 20.00 | 0.38 | TDF/TAF + RAL | 2 October 2019 | No |

LTFU: loss to follow up; HE: heterosexual; MSM: man who has sex with man; IDU: injective drug user; PjP: *Pneumocystis jirovecii* Pneumoniae; HZV: Herpes Zoster Virus; HSV: Herpes Simplex Virus; CMV: Cytomegalovirus; TDF: Tenofovir Disoproxil Fumarate; FTC: Emtricitabine; ATV/r: Atazanavir/ritonavir; TAF: Tenofovir Alafenamide Fumarate; DTG: Dolutegravir; RAL: Raltegravir; DRV/r: Darunavir/ritonavir.

## 4. Discussion

The hospital prevalence of HIV infections in the study population was surprisingly high, considering the small number of individuals tested. During the same year, in Italy, 2847 new HIV diagnoses were reported, with 48 in Sardinia. However, the total number of screenings performed is not available [19]. Of 48 new diagnoses, 11 have been discovered thanks to our intervention. This confirms that proactive testing and screening implementation are necessary to reach at least "the first 95%" of UNAIDS, and hopefully, zero new infections.

In Italy, an "opt-in" strategy is currently applied. This means that patients are asked if they want to undergo the HIV screening and must sign an informed consent. However, given the results of our study, an "opt-out" strategy in which the clinician can perform the test, if considered appropriate, would be recommended. Patients should refuse to undergo the test explicitly, as for almost all another routine testing. This approach would permit an increase in the number of PWH knowing their serological status and would contribute to normalizing the HIV infection and reducing stigma and self-stigma [18].

In a recent study by Garcia et al., the training of non-HIV specialists in a tertiary hospital in Spain has changed the attitudes toward screening and the number of new infections diagnosed. In particular, it led to a 24% increase in HIV tests requested after training (significant in medical and surgical departments (mainly gynecology)) and a 48% increase in HIV diagnosis after the training [20]. In addition, a recently published systematic review reported medical staff education as an effective method to improve HIV screening. In addition, rapid test provision and integrated routine testing, decision making tools, and campaigns targeting patients showed promising results [21].

Among the patients diagnosed, five have been tested for conditions associated with an undiagnosed HIV prevalence of at least 0.1%; thus, they would not have been tested aside from the project. Of note, 10 out of 11 (90.1%) patients reached undetectability and consequent immune recovery. It has been demonstrated that starting and adhering to combination ART (cART) has clinical benefits in reducing HIV-related morbidity and mortality and the risk of HIV transmission [8]. If on cART, people enrolled in the ICONA Foundation Study have been shown to maintain a stable HIV-RNA < 200 cp/mL, a condition named "Undetectable status" for almost 97% of the time during ten years of follow-up [22]. Thus, the U = U message strengthens the value of the SHOT Project, because all these people that have been diagnosed and adhere to treatment cannot transmit the virus. Consequently, we potentially avoided new cases of HIV infection.

A proactive indicator disease-guided screening can help avoid missed opportunities to diagnose the infection in a hospital setting. The feasibility and efficacy of an indicator conditions-based strategy were evaluated during the HIDES I study with encouraging results regarding early diagnosis [23]. The importance of this approach was confirmed by an audit conducted in 2013, which revealed more than 100 missed diagnosing opportunities based only on indicator conditions [24]. In addition, other interventions conducted in Amsterdam and Barcelona demonstrated significant efficacy from incentives such as audit or electronic prompt alerts, which may encourage and remind physicians which are the indicator conditions and when to offer an HIV screening [25,26]. Tincati et al., during the 17th AIDS Conference in Basel, presented a similar study, including 462 subjects, and they found an HIV prevalence of 2.2% [27].

Regarding the first-line treatment, all people started with a three-drug regimen. According to national and international guidelines [28,29], in all patients, the backbone was tenofovir disoproxil (or alafenamide), fumarate, and emtricitabine. Regarding the anchor drug, 9 out of 11 people started with an integrase inhibitor according to national and international guidelines. In particular, six patients started with dolutegravir 50 mg, and three with raltegravir 600 mg. The other two people started with a booster protease inhibitor, in one case with darunavir/cobicistat, and in the other case atazanavir/ritonavir.

In all cases (except for the woman who died) HIV undetectability was achieved, confirming the high efficacy of the antiretroviral treatment, including in people with a severe immunological deficiency [30–32].

In conclusion, although most of our patients were diagnosed with advanced stage, implementing this kind of intervention with community-based interventions could also improve early diagnosis.

In this direction, we advocate introducing HIV screening as part of good clinical practice, which will be possible only by raising awareness among healthcare providers.

**Author Contributions:** Conceptualization, G.M., P.B. and M.S.M.; methodology, A.D.V., G.M. and A.C.; validation, A.D.V. and A.C.; formal analysis, A.D.V., A.C., V.F. and M.S.M.; investigation, V.F., C.F., M.A.M., A.G.F., F.F., N.M. and C.P.; resources, P.B., C.F., M.A.M. and F.F.; data curation, M.S.M., A.G.F., N.M., C.P. and S.B.; writing—original draft preparation, A.D.V., A.C. and G.M.; writing—review and editing, V.F., M.S.M., P.B., C.F., M.A.M., A.G.F., F.F., N.M., C.P. and S.B.; supervision, S.B. and G.M. All authors have read and agreed to the published version of the manuscript.

**Funding:** This research received no external funding.

**Institutional Review Board Statement:** The study was conducted in accordance with the Helsinki Declaration approved in 2003, and the local ethic committee approved the protocol (Protocol 248I/CE, 11 April 2017).

**Informed Consent Statement:** Written informed consent was obtained from all subjects involved in the study.

**Data Availability Statement:** The data that support the findings of this study are available from the corresponding author, upon reasonable request.

**Acknowledgments:** We thank all the physicians and nurses involved in the SHOT program at the University Hospital of Sassari. We thank Gilead Sciences for supporting the publication.

**Conflicts of Interest:** The authors declare no conflict of interest.

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
