# Peer review of "HIV Infection Indicator Disease-Based Active Case Finding in a University Hospital: Results from the SHOT Project"

_2036-7449, doi:10.3390/idr15010010_

Round 1

Reviewer 1 Report

This is an interesting study on in-hospital indicator-conditions based HIV diagnosis. The topic is interesting and the study is well conducted; data are sound and the conclusions seems supported by the available data.

I have a few minor comments:

a. abstract: I would increase the methods part (skipping the study objectives) and reducing the discussion (the last three lines are probably too much for an abstract)

b. abstract: I would not start with there Italian situation... what b out something more general on the need for enhanced screening for attaining WHO 95/95/095 targets?

c. Background: what about mentioning the data on non-ID physicians' knowledge and attitude towards HIV?

d. Some typos are presented t in Figure 2 axis

e. I was not able to see Table 1

f. Which was the prevalence in non ID wards (usually those an increase risk of low testing?

g. The Discussion is balanced but probably more references to other similar projects and the EU survey on HIV testing strategies may complete it. I would also refer to the benefit of training non ID physicians...

Author Response

Dear Reviewer,

Thank you for reading our paper and giving us your precious suggestion to improve our work's quality.

Here is our reply point-by-point to your comments.

Reviewer (R): This is an interesting study on in-hospital indicator-conditions based HIV diagnosis. The topic is interesting and the study is well conducted; data are sound and the conclusions seems supported by the available data.

Authors’ reply (AR): Thank you for carefully reading our paper and for your comments.

R: abstract: I would increase the methods part (skipping the study objectives) and reducing the discussion (the last three lines are probably too much for an abstract)

AR: Thank you for your suggestion. The Abstract was modified accordingly.

R: abstract: I would not start with there Italian situation... what b out something more general on the need for enhanced screening for attaining WHO 95/95/095 targets?

AR: We decided to leave the part regarding Italian epidemiology since our project is based in Italy. However, accordingly to your suggestion, we included a mention to the three 95 targets by UNAIDS.

R: Background: what about mentioning the data on non-ID physicians' knowledge and attitude towards HIV?

AR: Thank you for your precious suggestion, a few lines were added.

R: Some typos are presented t in Figure 2 axis

AR: Thank you for having read our manuscript carefully. We have fixed the typos in Figure 2.

R: I was not able to see Table 1

AR: we are so sorry about it. We have uploaded it as a separate file. We have now added it to the manuscript.

R: Which was the prevalence in non ID wards (usually those an increase risk of low testing?

AR: Positivity rates were reported in the Results

R: The Discussion is balanced but probably more references to other similar projects and the EU survey on HIV testing strategies may complete it. I would also refer to the benefit of training non ID physicians...

AR: Thank you for your comment. We added a paragraph regarding experiences based on the same model.

Reviewer 2 Report

In this prospective observational single center study in Italy, the study authors report on their experience with a stratified screening algorithm for HIV in hospitalized patients. They perform a descriptive analysis and reveal new HIV diagnosis of 3.7% during their study period. Through this study, they aim to highlight the importance of more efficient screening to ensure better access and linkage to care.

Major Comments

1. The main comment that I have is that Table 1 (that is referenced on page 5 of 7) appears to be missing in the main text of the manuscript. Without this table, it is unclear what datapoints have been reported on by the study authors. I would request the study authors to please look into including this table – it is feasible it may not have been uploaded correctly during the manuscript submission.

2. Adding on to the above point, it would be helpful to know what antiretroviral regimens were these patients started on. As the study period was between 2017 and 2018, there have been newer combination antiretroviral therapies included.

3. The study authors report that their prevalence of 3.7% was “surprisingly high, considering the small number of individuals tested”. How does this compare to the rate of newly diagnosed HIV infections in Italy nationally or in the same state? Are there any similar studies that have a lower prevalence rate but performed with the same population numbers? Some recent studies (such as those by Young et al doi:10.1089/AID.2022.0054 and Zhao et al doi:10.7189/jogh.12.11015) report a lower prevalence rate than the current study, but involve larger patient populations over a longer period of time.

Minor Comments

1. In the abstract (page 1 of 7, line 21) and results (page 3 of 7, line 111), the patient characteristics are reported as “males (67,3%), and 98 were females (32,7%)”. I believe it should 67.3% and 32.7% respectively.

2. In Figure 2 (page 4 of 7), the spelling of Non-Hodgkin Lymphoma appears to be incorrect

3. Figure 2 appears to be missing a Legend that describes the abbreviations utilized (such as STI, CAP)

4. Additionally, in Figure 2, there are certain datapoints in the X-axis (such as the point related to Leukopenia/Thrombocytopenia and the point related to what I presume is Thrombotic thrombocytopenic purpura) that are missing part of their text.

Author Response

Dear Reviewer,

Thank you for reading our paper and giving us your precious suggestion to improve our work's quality.

Here is our reply point-by-point to your comments.

R: The main comment that I have is that Table 1 (that is referenced on page 5 of 7) appears to be missing in the main text of the manuscript. Without this table, it is unclear what datapoints have been reported on by the study authors. I would request the study authors to please look into including this table – it is feasible it may not have been uploaded correctly during the manuscript submission.

AR: Thank you for your comment. Unfortunately, there was a problem with file uploading. Table 1 has now been placed in the text.

R: Adding on to the above point, it would be helpful to know what antiretroviral regimens were these patients started on. As the study period was between 2017 and 2018, there have been newer combination antiretroviral therapies included.

AR: We agree with you. The antiretroviral regimens are described in Table 1.

R: The study authors report that their prevalence of 3.7% was “surprisingly high, considering the small number of individuals tested”. How does this compare to the rate of newly diagnosed HIV infections in Italy nationally or in the same state?

AR: Thank you for your comment. We have a sentence to explain better why we defined the 3.7% as “surprisingly high”. Furthermore, we have added data to a similar Italian study performed in another region with a lower prevalence of HIV positivity.

Are there any similar studies that have a lower prevalence rate but performed with the same population numbers? Some recent studies (such as those by Young et al doi:10.1089/AID.2022.0054 and Zhao et al doi:10.7189/jogh.12.11015) report a lower prevalence rate than the current study, but involve larger patient populations over a longer period of time.

AR: Thanks for you suggestion, we have added a paragraph discussing similar studies conducted in Europe and Italy (Lombardy).

Minor Comments

R: In the abstract (page 1 of 7, line 21) and results (page 3 of 7, line 111), the patient characteristics are reported as “males (67,3%), and 98 were females (32,7%)”. I believe it should 67.3% and 32.7% respectively.

AR: Thank you for your comment, corrections have been applied.

R: . In Figure 2 (page 4 of 7), the spelling of Non-Hodgkin Lymphoma appears to be incorrect

Thank you for your comment, corrections have been provided.

R: Figure 2 appears to be missing a Legend that describes the abbreviations utilized (such as STI, CAP)

AR: Thank you for your suggestion, we have provided a Legend

R:. Additionally, in Figure 2, there are certain datapoints in the X-axis (such as the point related to Leukopenia/Thrombocytopenia and the point related to what I presume is Thrombotic thrombocytopenic purpura) that are missing part of their text.

AR: Thank you, we have corrected datapoints.

Round 2

Reviewer 2 Report

Overall, the study authors have commendably incorporated most of the comments/suggestions. I have no major comments to add, except for the fact that part of the Table appears to be cut off in the draft that has been included. I would recommend that the authors work closely with the typesetting editors to ensure that the table is correctly displayed in the final version.

Apart from that, I just have a few minor comments as below.

Minor Comments

1. Page 2 of 10, lines 77-79 – the study authors note “A recently published study from Australia highlighted perceived stigma, testing and results responsibility, and limited resources as the main reasons for testing hesitancy”. Please provide the reference of this study.

2. Table 1, Page 6 of 10 – for patient 2, I believe the authors meant the reason for testing as “Thrush” instead of “Rush”.

3.  Table 1, Page 6 of 10 – please include the units of measurement for the HIV viral load.

4. Table 1, Page 6 of 10 – please indicate which patient in the table died (I believe it appears as patient #10).

Author Response

Dear author,

Thank you for your precious comments. Regarding Table 1, we were forced to add it as a figure. We'll send the word file to the editor to add it with the best layout.

R:  Page 2 of 10, lines 77-79 – the study authors note “A recently published study from Australia highlighted perceived stigma, testing and results responsibility, and limited resources as the main reasons for testing hesitancy”. Please provide the reference of this study.

AR: Thank you for your comment. We provided the reference.

R: 2. Table 1, Page 6 of 10 – for patient 2, I believe the authors meant the reason for testing as “Thrush” instead of “Rush”.

AR: Thank you for noticing this typo. We meant cutaneous "Rash". Therefore, we add the correct Table version.

R: Table 1, Page 6 of 10 – please include the units of measurement for the HIV viral load.

AR: thank you for your comment. We add the units of measurement.

R: Table 1, Page 6 of 10 – please indicate which patient in the table died (I believe it appears as patient #10).

AR: We indicated the patients who died; it was the number 4!